# Modelling Asymmetric Unemployment Dynamics: The Logarithmic-Harmonic Potential Approach

**DOI:** 10.3390/e24030400

**Published:** 2022-03-13

**Authors:** Cho-Hoi Hui, Chi-Fai Lo, Ho-Yan Ip

**Affiliations:** 1Hong Kong Monetary Authority, 55/F, Two International Finance Centre, 8, Finance Street, Central, Hong Kong 999077, China; 2Institute of Theoretical Physics and Department of Physics, The Chinese University of Hong Kong, Shatin, N.T., Hong Kong 999077, China; cflo@phy.cuhk.edu.hk (C.-F.L.); ian_1996426@hotmail.com (H.-Y.I.)

**Keywords:** logarithmic potential, quasibounded process, stochastic heat engines, unemployment rates

## Abstract

Asymmetric behaviour has been documented in unemployment rates which increase quickly in recessions but decline relatively slowly during expansions. To model such asymmetric dynamics, this paper provides a rigorous derivation of the asymmetric mean-reverting fundamental dynamics governing the unemployment rate based on a model of a simple labour supply and demand (fundamental) relationship, and shows that the fundamental dynamics is a unique choice following the Rayleigh process. By analogy, such a fundamental can be considered as a one-dimensional overdamped Brownian particle moving in a logarithmic–harmonic potential well, and a simple prototype of stochastic heat engines. The solution of the model equation illustrates that the unemployment rate rises faster with more flattened potential well of the fundamental, more ample labour supply, and less anchored expectation of the unemployment rate, suggesting asymmetric unemployment rate dynamics in recessions and expansions. We perform explicit calibration of both the unemployment rate and fundamental dynamics, confirming the validity of our model for the fundamental dynamics.

## 1. Introduction

Asymmetric behaviour has been documented in postwar US unemployment rates, suggesting that they increase quickly in recessions but decline slowly during expansions. Consider a scenario in which the economy is in a recession: in response to this negative shock, job vacancies (demand for labour) drop on impact, leading to a rise in unemployment. Then, consider the response to an equally-sized positive shock. While the adjustment in response to such positive shock is qualitatively similar, because of the different dynamics of demand for labour under the shocks, the unemployment rate increases by more in response to a negative shock than it drops in response to a positive one. This generates asymmetry between peaks and troughs of the unemployment rate as shown in Figure 1 with observations during 1948–2020. The importance of this issue has been stressed by many papers in the both theoretical and empirical literature [1,2,3,4,5,6,7].

To model unemployment, the workhorse Diamond–Mortensen–Pissarides (DMP) search model is an example of a theory model in which employment becomes stuck in an equilibrium under shocks to the economy without maximising employment and consumption [3,8]. Its building blocks include: a stochastic model of labour turnover with workers being unemployed, and then finding new jobs; a model of labour-market tightness; and a bargaining model of wage determination capturing workers’ productivity and their wages. However, this model has a well-known but counter-intuitive property of lacking feedback from unemployment to labour-market tightness. This suggests that jobs are just as easy to find and the situations are not affected by the levels of the unemployment rates. Therefore, the unemployment rate dynamics in the DMP model do not show any asymmetry during economic contractions and expansions.

Much of the available research is devoted to studying modifications of the DMP model to alter its counter-intuitive property and explaining the asymmetric dynamics of unemployment rates. To address the asymmetric dynamics, models are considered with downward nominal wage rigidity [9,10]. Other mechanisms that address the gradual reduction of unemployment during a recovery include: a gradual return to the normal mix of unemployment; slowness in labour-market recovery following a recession; and congestion effects impeding recruitment efforts when unemployment is high [7,11,12].

To investigate the asymmetry of unemployment rates during economic contractions and expansions, this paper develops an equilibrium model based upon a simple labour supply and demand (fundamental) relationship. The model explicitly derives the stochastic processes of the unemployment rate from the associated dynamics of the fundamental (demand for labour and government interventions), and their linkage generating the asymmetry. Different from the DMP model and its modifications using a system of models for the key factors, in this model, the behaviour of the labour market and government interventions form an aggregate fundamental following a stochastic process under a logarithmic–harmonic potential. The potential measures labour-market tightness, government interventions (if any), and other economic factors affecting employment. The proposed model provides an intuitive explanation for the asymmetry of unemployment rates which rise faster with the flattened logarithmic–harmonic potential of the fundamental, i.e., less self-restoring labour demand and less effective government interventions, more ample labour supply, and less anchored expectation of unemployment during economic contractions.

To model the asymmetric unemployment rate dynamics, this paper shows that the fundamental dynamics governing the unemployment rate can be derived from a second-order ordinary differential equation based upon a simple labour supply and demand (fundamental) relationship. By imposing a reflecting (so-called smooth-pasting) boundary condition at an upper threshold for the equation, the fundamental dynamics described by the Rayleigh process which forms a logarithmic–harmonic potential is shown to be the unique choice. The Rayleigh process represents the self-correcting process in the labour market and government interventions. The corresponding unemployment rate follows a quasibounded process such that it can breach the threshold if the probability leakage condition is met under a severe negative-demand shock. We perform explicit calibration of both the unemployment rate and fundamental dynamics, confirming the validity of our model of the fundamental dynamics. The model explains the empirical observations that the unemployment rates move asymmetrically in recessions and expansions.

This paper is organised as follows. In the next section, we develop the model from log-linear and stochastic differential equations based upon a relationship between labour supply and demand, and a logarithmic–harmonic potential for the fundamental dynamics of the unemployment rate. The corresponding solutions of the model equations are derived and analysed with the empirical calibration of the model in the third section. The final section concludes the paper.

## 2. Methods

### 2.1. Proposed Model

The unemployment rate is defined as the percentage of unemployed workers in the total labour force. Workers are considered unemployed if they currently do not work, but are looking for jobs. The total labour force consists of all employed and unemployed people in a labour market. By keeping other factors constant, an increase in the labour supply or a decrease in labour demand will increase the number of unemployed workers. Therefore, the unemployment rate is directly proportional to the labour supply but inversely proportional to the labour demand. Such relationship for the unemployment rate *R* can be expressed as:(1)$$R=k\left(\frac{L}{Q}\right);\;k\;>\;0$$
where *L* is the labour supply, *Q* is the labour demand and *k* is the proportionality constant which measures the response of *R* to a change in the ratio (*L*/*Q*).

Under equilibrium, Equation (1) has a corresponding equation:(2)$$\bar{R}=k\left(\bar{\frac{L}{\bar{Q}}}\right),$$
where $\bar{R},$ $\bar{L}$ and $\bar{Q}$ are the equilibrium values of *R*, *L* and *Q* respectively. By combining Equations (1) and (2) and incorporating market expectations of the unemployment rate, we obtain the following log-linear equation:(3)$$r=l+\nu +\alpha \frac{E\left[dr\right]}{dt},$$
where *r*, *l* and *ν* are the normalised log *R*, *L* and *Q* against their equilibrium values given as:(4)$$r=\ln\left(\frac{R}{\bar{R}}\right);$$
(5)$$l=\ln\left(\frac{L}{\bar{L}}\right);$$
(6)$$\nu =-\ln\left(\frac{Q}{\bar{Q}}\right),$$
*α* is the absolute value of semielasticity of the unemployment rate with respect to its expected rate of change, and *E* the expectation operator. The last term in Equation (3) represents the expected change in the labour market. The log-linear equation is used for the standard flexible-price monetary model to study the exchange rate dynamics. The equation is based on the existence of a money demand function, the purchasing power parity, and the uncovered interest rate parity [13,14,15].

We assume the labour supply is constant given that the total labour force does not change much over time. The demand shock (“fundamental”) (*ν*) follows a stochastic process with a drift $\mu_{\nu }$ which is a function of *ν* and instantaneous standard deviation $\sigma_{\nu }$:(7)$$d\nu =\mu_{\nu }dt+\sigma_{\nu }dZ,\;\mathrm{for}\;\nu \in \left(-\infty ,0\right]$$
where *dZ* is a Wiener process with E[dZ]=0 and $E\left[dZ^{2}\right]=dt$.

Ito’s lemma is applied to Equations (3) and (7) with some algebra to obtain a 2nd-order linear ordinary differential equation:(8)$$\frac{1}{2}\alpha \sigma_{\nu }^{2}\frac{d^{2}r}{d\nu^{2}}+\alpha \mu_{\nu }\frac{dr}{d\nu }-r=-\nu -l$$

If a government does not intervene in the economy to offset demand shocks to the fundamental and is expected to remain passive whatever the employment rate moves, the driving process of the fundamental is simple, with a zero trend of $\mu_{\nu }=0$. The solution of Equation (8) is:(9)r=ν+l.
On the other hand, the government may intervene at certain unemployment rates to influence the labour market by altering the stochastic process governing the labour demand driving the fundamental *ν*. In addition, there is self-adjustment in the labour market such as an increase in the demand with lower labour cost. Therefore, the stochastic fundamental no longer follows a zero trend $\mu_{\nu }$ and thus the solution in Equation (9) is invalid. The solution needs to take into account both the fundamental’s dynamics associated with the interventions and self-adjustment, and the boundary condition.

Given that the government will intervene when the unemployment rate rises toward to a threshold $\hat{r}$, we impose the following boundary conditions at the fundamental ν=0 for Equation (8):(10)$$r\left(\nu =0\right)=\hat{r};$$
(11)$${\frac{dr\left(\nu \right)}{d\nu }|}_{\nu =0}=0,$$
where the condition of Equation (10) ensures a proper normalisation of the unemployment rate, and Equation (11) is the smooth-pasting boundary condition at ν=0. The smooth-pasting condition ensures the rate does not cross the threshold $\hat{r}$, suggesting an optimal boundary condition for the process. If the smooth-pasting condition breaks down, the rate could jump across the threshold. Hence, the break-down condition at the boundary allows the stochastic process of the fundamental to be quasibounded as shown in Section 3. The quasibounded stochastic process has applications in finance, exchange-rate systems and various systems with spatial confinement [16,17,18,19,20,21,22,23,24]. It is also applied to model the tumour cell growth [25].

### 2.2. Logarithmic-Harmonic Potential

The most general form of *µ* can be expressed as
(12)$$\mu =\overset{\infty }{\underset{n=-\infty }{\sum }}A_{n}\nu^{n},$$
where the coefficients $\left\{A_{n}\right\}$ are arbitrary real constants. The assumption of the differential equation Equation (8) having no irregular singular point dictates that $A_{n}=0$ for n<−1. This is justified because a solution near an irregular singular point has rather extreme behaviours; it may blow up exponentially, vanish exponentially, or oscillate wildly. The coefficient $A_{-1}$ must be positive in order that the singular drift component $A_{-1}\nu^{-1}$ prevents ν from breaching the boundary at ν=0. On the other hand, a nonpositive $A_{-1}$ makes the boundary at ν=0 no longer impenetrable. In addition, to ensure that the boundary at ν→−∞ is inaccessible, μ must be positive in this asymptotic limit of ν. Beyond question, the simplest possible candidate of this class of *µ* can be obtained by setting $A_{n}=0$ for n>1, $A_{1}<0$ and $A_{-1}>0$. The mean-reverting component $A_{1}\nu$ clearly pulls ν away from the limit at ν →−∞. However, the constant drift term $A_{0}$ has a conflicting role: a negative $A_{0}$ reinforces the singular barrier at ν=0 and weakens the mean-reversion, whereas a positive $A_{0}$ has the opposite effect. It is thus natural to have a vanishing $A_{0}$ in μ. As a result, the asymmetric mean-reverting μ, which turns out to be the unique choice [14,15]: (13)$$\mu =\frac{A_{-1}}{\nu }+A_{1}\nu ,$$
For $A_{1}<0$ and $A_{-1}>0$. The corresponding stochastic process is commonly known as the Rayleigh process [26]. The coefficient $A_{-1}$ plays the critical role of determining the fundamental dynamics and corresponding unemployment rate. It is clear the special case of vanishing $A_{1}$ and $A_{-1}$ indicates the absence of government interventions or self-correction in the labour market.

To better understand the fundamental dynamics, we draw an analogy between the fundamental and a one-dimensional overdamped random particle in the presence of an external conservative force [27]. Whilst the fundamental *ν* is governed by the stochastic differential equation in Equation (7), the position variable ξ of the random particle (fundamental) obeys the equation (in appropriate units):(14)$$d\xi =F\left(\xi \right)dt+\sqrt{2D}dZ,$$
where D is the diffusion coefficient and F(ξ)≡−dU(ξ)/dξ is the external force with U(ξ) being the corresponding potential well. The fundamental (particle) ν follows a stochastic process subject to an external force defined by
(15)$$F\left(\nu \right)=\frac{A_{-1}}{\nu }+A_{1}\nu .$$
Then the transition probability density for the position of the particle, Y(ν,t), can be described by the Fokker–Planck equation:(16)$$\frac{\partial }{\partial t}Y\left(\nu ,t\right)=-\left\{D\frac{\partial^{2}}{\partial \nu^{2}}-\frac{\partial }{\partial \nu }\left[\frac{A_{-1}}{\nu }+A_{1}\nu \right]\right\}Y\left(\nu ,t\right).$$
By direct integration over ν, the corresponding logarithmic–harmonic potential well U(ν) can be determined as:(17)$$U\left(\nu \right)=-\int^{\;}\left(\frac{A_{-1}}{\nu }+A_{1}\nu \right)d\nu =-A_{-1}\ln\left|\nu \right|-\frac{1}{2}A_{1}\nu^{2}.$$

The logarithmic potential is applied to a large variety of problems in chemical, statistical, and biological physics. For example, the Brownian particle in a logarithm potential can represent a line of charges, which is used to model the interactions of colloids and polymers with walls of narrow channels and pores [28,29]. In generating denaturation bubbles of double-stranded DNA, the logarithmic potential is an entropic term in the free energy cost of unzipping DNA base-pairs [30,31].

## 3. Results

### 3.1. Asymmetric Unemployment Rates

As shown in Figure 2 for different $A_{-1}$ and $A_{1}$, this logarithmic–harmonic potential forms an asymmetric and anharmonic potential well over the semi-infinite interval (−∞,0] in which the particle moves about randomly [32,33,34,35,36]. The shapes of the potential well reflect the self-adjustment of the labour market and the government’s capability to intervene the market as an external force. Decreasing the magnitude of $A_{1}$ will give an extremely flat potential well covering almost the entire semi-infinite range of ν such that the Brownian force drives the fundamental’s motion. A weak mean-reverting force allows the fundamental to move more randomly. Similarly, decreasing $A_{-1}$ will allow the fundamental to approach the boundary at ν=0 more easily and increase the probability of ν breaching the boundary; a nonpositive $A_{-1}$ will even make the potential well no longer bounded at ν=0. Hence, the stochastic process of the fundamental is actually quasibounded, suggesting the existence of a collapse of the labour market during a severe recession and possible break-down of the smooth-pasting condition at the boundary with the unemployment rate breaching the threshold at $\hat{r}$.

The model is closed by specifying the intervention policy at the upper boundary. Instead of assuming the government to intervene to keep the unemployment rate below a fixed rate at all times, we assume that the interventions are kept only a moving average R of past rates. Therefore, the government is mindful of unemployment rate movements over a time interval, instead of its current level. The upper boundary is considered as a limit for a distribution of the rate’s mean and standard deviation. Without assuming any distribution of the rate *R*, the upper boundary $R_{U}$ is taken to be the number (∆) of standard deviations (Σ) from its mean $\bar{R}$: $R_{U}=\bar{R}+∆\Sigma$. The level of the upper boundary is set to be adequately high. Given a normal distribution with ∆ equal to 1.5 and 2, the corresponding percentage increases from the mean are 37.5% and 50% respectively; and the cumulative normal probabilities above the boundary are 0.0668 and 0.022. The method to measure the historical rates and the choice of the level of the boundary do not affect the derivation of the solution and the qualitative results. For this reason, the use of the moving average $R_{At}$ seems both simple and reasonably realistic. $R_{At}$ represents the equilibrium unemployment rate $\bar{R}$ in Equation (4). For sustaining unemployment pressure, the moving average can be scaled by a parameter $\eta_{U}>1$, such that $\eta_{U}R_{At}$ forms an upper boundary. The parameter $\eta_{U}$ tells how far the unemployment rate can rise. With no loss of generality, the normalised log unemployment rate *r* is redefined by:(18)$$r=-\ln\left[\frac{R}{\eta_{U}R_{At}}\right];\;0\leq r<\infty ,$$
where *r* = 0 is the corresponding boundary of $R=\eta_{U}R_{At}$ scaled by a moving average of rates over a time horizon. By normalizing the unemployment rate with a moving boundary, the relationship between the unemployment rate and the associated fundamental depends upon the historical rate.

By the power series method, the solution of Equation (8) is obtained as:(19)$$r\left(\nu \right)=\nu^{2}\sum_{n=0}^{\infty }B_{n}\nu^{n}.$$
This series solution is a convergent series for all *v* according to the ratio test [21,22,23,24]. Given the rapid convergence of the series solution, we propose to approximate the exact solution by an optimal approximate solution of the form:(20)$$r\left(\nu \right)\approx \varepsilon B_{0}\nu^{2}=-\frac{\epsilon l}{\alpha \left(\sigma_{\nu }^{2}+2A_{-1}\right)}\nu^{2}$$
where ε is a positive parameter determined by minimising the total error between the approximate solution and the power series solution.

Figure 3 plots the relationship between the unemployment rate in the original measure *R* and the fundamental *ν* expressed in Equation (20) based on the empirical estimations in Figure 4. It shows that changes in the rate at the upper boundary flatten with changes in the fundamental. This suggests that even when the fundamental changes materially, the unemployment rate only marginally moves away from the upper boundary. When a negative-demand shock pushes the rate towards the boundary and *ν* towards zero, there is a force to pull them back acting as a stabilising force which is the term $\left(\frac{A_{-1}}{\nu }\right)$ in Equation (15) to limit a further increase in the rate. A factor behind the restoring force is the government’s economic stimulus policy to increase the labour demand. According to the model, given that changes in the demand alter the fundamental dynamics, the unemployment rate could move between D and A; or D and C, where the paths depend on the coefficient $\varepsilon B_{0}$ in Equation (20). $\varepsilon B_{0}$ reflects the labour-market condition determined by the parameters ($A_{-1}\;\mathrm{and}\;A_{1}$) of the logarithmic–harmonic potential of the fundamental, the labour supply (*l*), and sensitivity (*α*) of the unemployment rate to its expected rate of change. A larger $\varepsilon B_{0}$ implies that the unemployment rate is more sensitive to changes in the fundamental (demand) with flattened logarithmic–harmonic potential well (small $A_{-1}\;\mathrm{and}\;A_{1}$) of the fundamental, ample labour supply *l*, and small *α*. This scenario of large $\varepsilon B_{0}$ happened during the recessions in the 1970s and after the 2008 global financial crisis, as shown in Figure 4 of the empirical estimations using the US unemployment rates. This suggests that the unemployment rate rises faster with flattened logarithmic–harmonic potential of the fundamental, more ample labour supply, and less anchored expectation of unemployment, as illustrated by the rate increasing from A to D with $\varepsilon B_{0}=0.9$. Conversely, during economic expansions, as in the early 1980s and 1990s, the rate decreases from D to C given the smaller $\varepsilon B_{0}=0.525$. The different values of $\varepsilon B_{0}$ demonstrate the asymmetric dynamics of unemployment rates during recessions and expansions.

### 3.2. Unemployment Rate Dynamics

To illustrate the unemployment rate dynamics, by applying Ito’s lemma to Equation (7) with Equation (20), *r* is shown to be governed by the following mean-reverting square-root process:(21)$$dr=\kappa \left(\theta -r\right)dt+\sigma_{r}\sqrt{r}dZ$$
where
(22)$$\kappa =2\left|A_{1}\right|,$$
(23)$$\theta =\varepsilon \left|\frac{B_{0}}{A_{1}}\right|\left(A_{-1}+\frac{1}{2}\sigma_{\nu }^{2}\right),$$
(24)$$\sigma_{r}=2\sigma_{\nu }\sqrt{\left|\varepsilon B_{0}\right|}.$$
*κ* determines the speed of the mean-reverting drift towards the long-term mean (equilibrium) *θ*.

Following Feller’s classification of boundary points, it can be inferred that the one at the origin is a boundary of no leakage for $(\sigma_{r}^{2}/4\kappa \theta )\leq 1$ in Equation (21), and it is not otherwise [37]. When the no-leakage condition holds, it prevents the unemployment rate *r* (*R*) from breaching the boundary at the origin $\left(\eta_{U}R_{At}\right)$; otherwise, the rate could pass through it. Therefore, *r* is quasibounded at the boundary. If the no-leakage condition does not hold, the smooth-pasting condition of Equation (11) at the boundary may break down. There is a nonattractive natural boundary (i.e., inaccessible) at infinity.

The probability density function (PDF) of *r* is given by:(25)$$G\left(r,t;r^{\prime },t^{\prime }\right)=\frac{2}{\sigma_{r}^{2}C_{1}\left(t-t^{\prime }\right)}{\left(\frac{r}{r^{\prime }}\right)}^{\frac{\omega }{2}}\exp\left[-\frac{\omega +2}{2}C_{2}\left(t-t^{\prime }\right)\right]\exp\left\{-\frac{2r^{\prime }+2r\exp\left[-C_{2}\left(t-t^{\prime }\right)\right]}{\sigma_{r}^{2}C_{1}\left(t-t^{\prime }\right)}\right\}\times I_{\omega }\left\{\frac{4r^{1/2}r^{\prime 1/2}\exp\left[-C_{2}\left(t-t^{\prime }\right)/2\right]}{\sigma_{r}^{2}C_{1}\left(t-t^{\prime }\right)}\right\},$$
where $\omega =2\kappa \theta /\sigma_{r}^{2}-1$, $C_{1}\left(\tau \right)=\left[\exp\left(\kappa \tau \right)-1\right]/\kappa$, $C_{2}\left(\tau \right)=-\kappa \tau$, and $I_{\omega }$ is the modified Bessel function of the first kind of order *ω*. It is not difficult to show that when $(\sigma_{r}^{2}/4\kappa \theta )>1$, we observe probability leakage through the boundary at the origin, implying that the probability of the unemployment rate breaching the upper limit is not zero. Conversely, for $(\sigma_{r}^{2}/4\kappa \theta )\leq 1$, there is no probability leakage and the total probability is preserved. With no probability leakage at the origin, the associated asymptotic PDF will ultimately converge to the steady-state distribution:(26)$$K\left(r,t\to \infty ;r^{\prime },t^{\prime }\right)=\frac{2r^{\omega +1/2}}{\Gamma \left(\omega +1\right)}{\left(\frac{2\kappa }{\sigma_{r}^{2}}\right)}^{\omega +1}\exp\left[-\frac{2\kappa }{\sigma_{r}^{2}}r\right],$$
where Γ denotes the gamma function.

Furthermore, by applying Ito’s lemma it can be shown that the normalised unemployment rate $\tilde{R}\equiv \exp\left(-r\right)$ obeys the stochastic differential equation:(27)$$d\tilde{R}=\tilde{\kappa }\left(\ln{\tilde{R}}_{\theta }-\ln R\right)Rdt+\sigma_{x}R\sqrt{-\ln R}dZ,$$
where ${\tilde{R}}_{\theta }=\exp\left(-\kappa \theta /\tilde{\kappa }\right)$ and $\tilde{\kappa }=\kappa +\sigma_{x}^{2}/2$. The variance $\sigma_{r}^{2}{\tilde{R}}^{2}\left(-\ln\tilde{R}\right)$ achieves its maximum at $\tilde{R}=\exp\left(-1/2\right)$ and vanishes at the boundary. Although in the model the variance of the unemployment rate declines towards the upper boundary, yet the rate could breach the quasibounded boundary under particular conditions. In addition to the asymmetry of the variance about the central parity, i.e., $\tilde{R}=0.5$, the deterministic drift term is manifestly asymmetrical about the central parity, too. In particular, in the neighbourhood of $\tilde{R}=0$ (lower limit), the drift term vanishes whereas near $\tilde{R}=1$ (upper limit), the restoring force remains finite. As a consequence, the asymmetry of the stochastic process of $\tilde{R}$ reflects the different nature of the upper and lower limits as well as the corresponding government intervention policy.

### 3.3. Model Validation

The parameters of the log-linear equation in Equation (20), i.e., $\varepsilon B_{0}$, are estimated by a simple procedure as follows. By substituting Equation (20) into Equation (3) yields
(28)$$r\left(t\right)=l-\sqrt{\frac{r\left(t\right)}{\varepsilon B_{0}}}+\alpha \frac{E\left[dr\right]}{dt},$$
From the time series of *r* we can construct the time series of both $\sqrt{r}$ and *dr*/*dt*. These two newly generated time series can be combined to form a new time series of χ, which is defined by the right-hand side of Equation (28). The parameter $\varepsilon B_{0}$ of the time series of can be determined by best fitting to the time series of *r*. The construction of the series $\frac{E\left[dr\right]}{dt}$ is done by using the 60-month moving average of *ds*. The estimations cover the monthly data of the US unemployment rates from 1948 to 2020 using a 20-year rolling window. Moreover, the model parameters of the dynamics of *r* in Equation (21) can be empirically calibrated by the maximum likelihood estimation for the log-likelihood function in Equation (25) with the associated moving boundary with $\eta_{U}=1.625$ (about 1.5 standard deviations (∆)) and a 12-month moving average.

The estimations for $\varepsilon B_{0},$ κ,  θ, and $\sigma_{r}$ shown in Figure 4 suggest that they are statistically significant in terms of their respective *t* and *z* statistics. Then, by combining Equations (20) and (22)–(24), which link up the parameters for the dynamics of the fundamental and unemployment rate, the model parameters *A*_−1_ and *A*_1_ in Equation (13) are obtained readily. Therefore, we are able to perform explicit calibration of the asymmetric fundamental shock, confirming the validity of our model of the fundamental dynamics.

Panel A of Figure 4 shows that large $\varepsilon B_{0}$ happened during the recessions in the 1970s and after the 2008 global financial crisis. Otherwise, smaller $\varepsilon B_{0}$ is estimated during economic expansions in the early 1980s and 1990s. Panel B shows that the estimated *κ* was significant with the *z*-statistic higher than the 10% significance level during most of the estimation period, reflecting significant restoring force in the unemployment rate dynamics towards its equilibrium level. However, *κ* fell sharply in the 2008 and was insignificant in a short period of time. The restoring force had diminished substantially with *κ* not different from zero statistically. Similar to *κ*, the estimated mean *θ* shown in Panel C was stable. Then *θ* fell in 2008 but remained statistically significant. The estimated volatility *σ_x_* shown in Panel D has been stable since the 1990s. The corresponding *z*-statistic is much higher than the 10% significance level, indicating that the estimated *σ_x_* is highly significant and the square-root-process part of the unemployment rate dynamics is robust.

### 3.4. Discussion

A simple logarithmic–harmonic potential approach has been presented for modelling the unemployment rates which are constrained to lie below an upper bound. The model is not only capable of capturing the empirical dynamics of the unemployment rates but also highly intuitive to explain its asymmetric dynamics. The derived stochastic process has a quasibounded upper limit, implying that the limit can be breached if the probability leakage condition is met. The quasiboundedness of the process at the upper boundary can thus provide us an indicator of possible severe downturns in the labour market. Empirical calibration of model parameters of the proposed process can also be easily performed due to the availability of an analytically tractable PDF. Hence, in terms of the calibrated model parameters, making predictions of future movements of unemployment rates become feasible. This is left for future research. However, given that $\varepsilon B_{0}$ is an aggregate coefficient of the parameter ($A_{-1})$ of the logarithmic potential, labour supply *l* and expectation *α* of the unemployment rate, further analysis of these economic factors could be limited in terms of their individual statistical estimations.

In addition, it has been pointed out that by analogy such a fundamental can be treated as a one-dimensional overdamped Brownian particle moving in a logarithmic–harmonic potential well. This random particle system has been intensively studied as a simple prototype of stochastic heat engines and may be realised in experiments [38,39,40]. Perhaps this prototype stochastic heat engine may help shed light on deciphering the secrets of other macroeconomic factors such as inflation rates, wages and exchange rates which also behave asymmetrically during different states of an economy.

## 4. Conclusions

We developed an equilibrium model in which the behaviour of the labour market and government interventions form an aggregate fundamental following the Rayleigh process in a logarithmic–harmonic potential. While the approach is different from conventional economic models, the logarithmic–harmonic potential captures labour-market tightness, government interventions, and other economic factors affecting employment. The solution of the model equation illustrates that the unemployment rate rises faster with a more flattened potential well of the fundamental, more ample labour supply, and less anchored expectation of the unemployment rate, suggesting asymmetric unemployment rate dynamics in recessions and expansions. The validity of the model is confirmed by the empirical calibration of both the unemployment rate and fundamental dynamics. This paper provides a rigorous derivation of fundamental dynamics for unemployment rates which are asymmetrically mean reverting. It also shows that the proposed fundamental dynamics is indeed the unique choice.

## Figures and Tables

**Figure 1 entropy-24-00400-f001:**
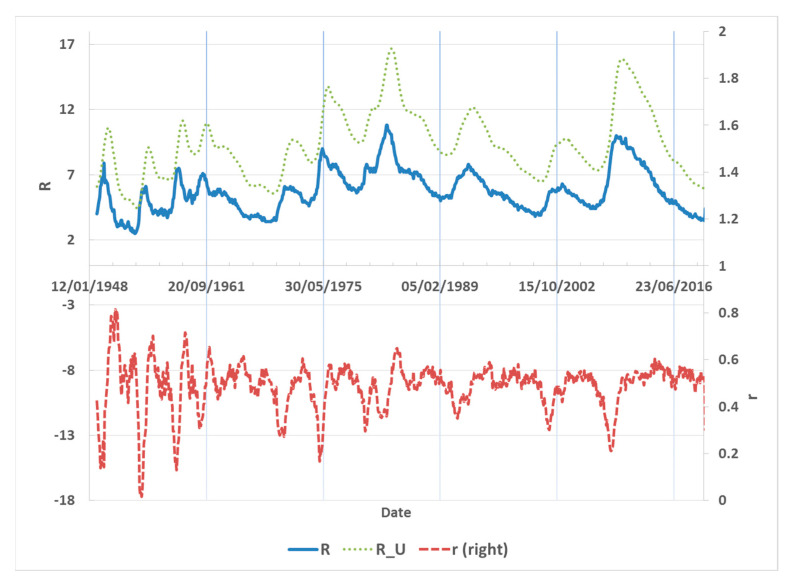
The US unemployment rate in *R*-scale (*left* axis) and *r*-scale (*left* axis), and upper boundary (R_U) in *R*. Source: Federal Reserve Bank of St. Louis.

**Figure 2 entropy-24-00400-f002:**
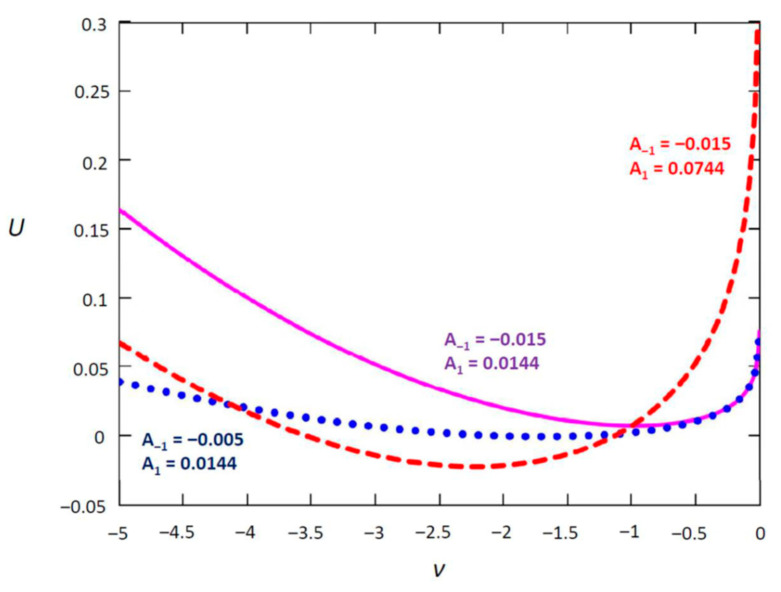
Potential U(ν) with different model parameters *A*_1_ and *A*_−1_.

**Figure 3 entropy-24-00400-f003:**
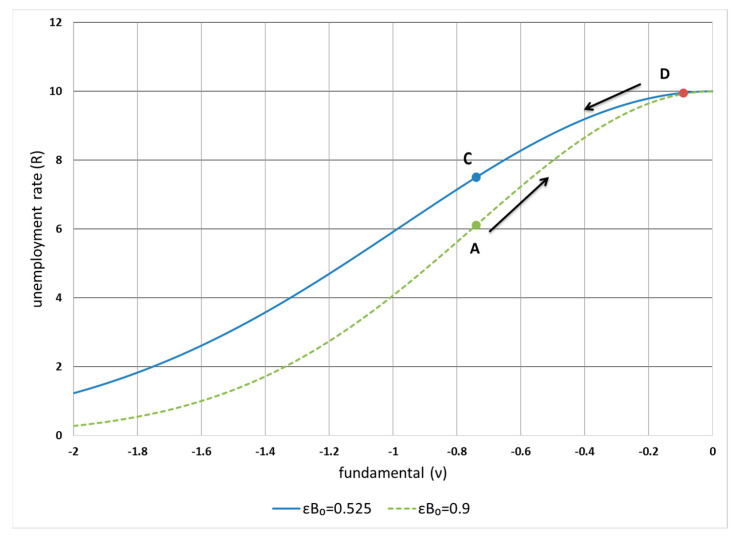
Relationship between unemployment rate (*R*) and fundamental (*ν*) based on Equation (20) with $\varepsilon B_{0}$ = 0.525 and 0.9.

**Figure 4 entropy-24-00400-f004:**
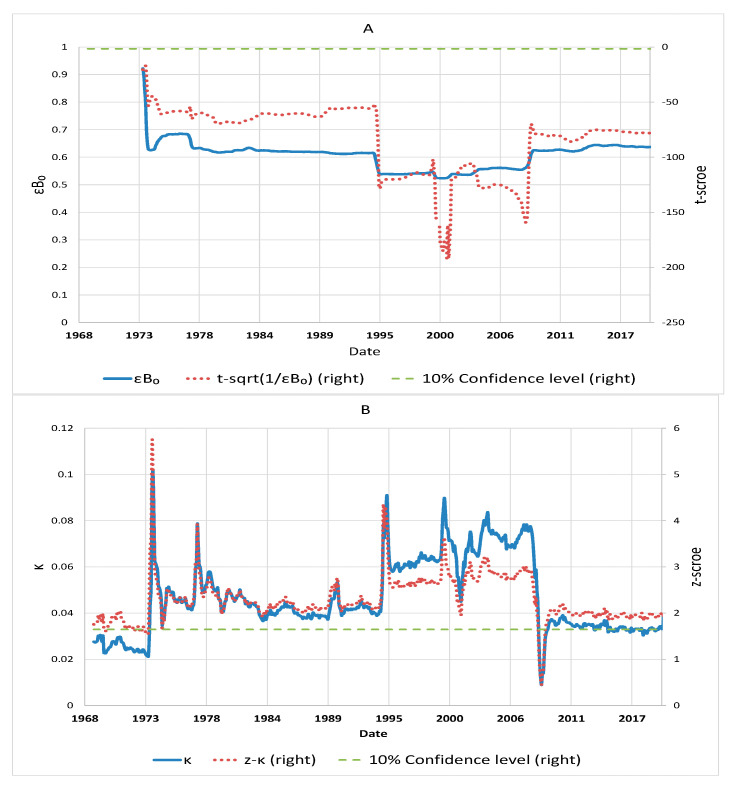
Estimated *εB*_0_ (Panel **A**), *κ* (Panel **B**), *θ* (Panel **C**), *σ_r_* (Panel **D**) and corresponding *t*- and *z*-statistics with moving boundary with 12-month moving average and 20-year rolling window.

## Data Availability

The US unemployment rate data is available from the Federal Reserve Bank of St. Louis at https://fred.stlouisfed.org/series/UNRATE, 14 February 2022.

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
