# Peer review of "Modelling Asymmetric Unemployment Dynamics: The Logarithmic-Harmonic Potential Approach"

_entropy, 2022, doi:10.3390/e24030400_

Round 1

Reviewer 1 Report

From the overall presentation I would say that interesting research work has been done. The topic is also important for the readers of the journal. However, I have a few more significant challenges with the paper. 

 Objectives and research questions are not clearly defined and should be included more clearly in the introduction section.  

The theoretical part remains at a modest level. At this stage, it does not yet provide an in-depth review of the previous literature. It is more a description than analysis. Therefore, a more detailed explanation of theoretical background and research design needs to be supplemented for this paper to be published. 

The research methods used are appropriate but have limitations, and this should be mentioned. The validation of the models could be presented and justified. Furthermore, the uncertainties of the applied analysis could be discussed. Finally, it would be appropriate to specify in more detail how this research differs from the already published paper that deals with a similar topic. 

The original contribution of the research has to be presented by focusing on the research results based on the research questions. 

You need to improve the practical and academic implications. 

However, the paper has to underline the limits of the research and future work. 

The authors have to pay attention to references inside the paper as well as the reference list. 

Reviewer 2 Report

Thank you for giving me the opportunity to review this manuscript. The idea of an asymmetric model for unemployment starting from the Brownian movement of the particles is impressive. But there are some missing parts in the paper:

  • the authors are stating that the unemployment is asymmetric, they should present a robust state of the arts with previous researches that had this conclusion;
  • identify the gap in the literature and setup the scope of the paper and the hypothesis;
  • the ”Methods” - should be split in Research methodology, Proposed model, Data and model validation, Results and findings;
  • the Discussions  has to be extended and do not rewrite the abstract, the authors should explain the how the model should be used, if there are any application of it;
  • a section of conclusions is needed to highlight the contribution of the authors to the theory and practice.
  • the references list has to be larger and to sustain the state of the arts, methodology, model, discussions and conclusions.

Reviewer 3 Report

The subject of the article is interesting and worth describing. Especially in the time of the economic crisis caused by COVID-19, this is a topical topic. However, the method of implementation is inadequate. In the Introduction, the authors presented an introduction to the subject. The Introduction section is deficient. It does not contain all the necessary elements. The main goal of the research and specific goals were not clearly defined. The authors vaguely provided a research gap. You need to clearly write down the goals just after specifying the research gap. In the Introduction section, the research hypotheses should be given. Alternatively, research questions may also be given.

The layout of the work is incorrect. I have already listed what should be in section 1 Introduction. The section 2 should be called Materials and Methods. The scope of the tests, the methods used and the test sequence should be given in turn. It is not transparent at the moment.

You have to think about the contents of section 2 on methods and section 3 results. This is debatable at the moment.

There is a Discussion section in the article, but it does not contain the required items. I understand a discussion as referring to other studies after presenting my research results. In my opinion, doing research without a clear comparison and reference to other research results in the fact that the obtained results cannot be properly assessed. There is a literature review, but it is very modest, because there is a link to three literature items at the end of the sentence. There is no broad scientific discussion.

There should also be a Conclusions Section. The authors mistakenly named the section as Discussion. There are required components that should appear in the Conclusions section. You must certainly refer to the hypotheses set at work. Have they been verified positively or negatively? Restrictions on the tests performed should be given. Topics for future research must also be provided.

There are also some minor remarks:

References to literature in the text were not properly made. There should be numbers in parentheses, e.g. [1} etc.

The X axis in figures 3 and 4 is not legible

The title of the figure should be below the figure. Not all figures have a title. If several figure are assigned to a given title, they must be numbered, e.g. a) b).

Round 2

Reviewer 1 Report

Dear Authors, 

In the revised version, the manuscript has been extended and improved. 

Best regards 

Reviewer 2 Report

The authors considered the suggestions and improved the manuscript accordingly.